# Multilevel modeling of county-level excessive alcohol use, rurality, and COVID-19 case fatality rates in the US

George Pro[1]*, Paul A. Gilbert[2], Julie A. Baldwin[3], Clare C. Brown[4], Sean Young[5], Nickolas Zaller[1]

1 Department of Health Behavior and Health Education, Fay W. Boozman College of Public Health, University of Arkansas for Medical Sciences, Little Rock, Arkansas, United States of America, 2 Department of Community and Behavioral Health, University of Iowa College of Public Health, Iowa City, Iowa, United States of America, 3 Center for Health Equity Research, Northern Arizona University, Flagstaff, Arizona, United States of America, 4 Department of Health Policy and Management, Fay W. Boozman College of Public Health, University of Arkansas for Medical Sciences, Little Rock, Arkansas, United States of America, 5 Department of Environmental and Occupational Health, Fay W. Boozman College of Public Health, University of Arkansas for Medical Sciences, Little Rock, Arkansas, United States of America

* gcpro@uams.edu

**Data Availability Statement:** The COVID-19 data used in this study is free and publicly available online through the New York Times COVID Data Repository (https://github.com/nytimes/covid-19-

## Abstract

### Objective

Reports of disparities in COVID-19 mortality rates are emerging in the public health literature as the pandemic continues to unfold. Alcohol misuse varies across the US and is related to poorer health and comorbidities that likely affect the severity of COVID-19 infection. High levels of pre-pandemic alcohol misuse in some counties may have set the stage for worse COVID-19 outcomes. Furthermore, this relationship may depend on how rural a county is, as access to healthcare in rural communities has lagged behind more urban areas. The objective of this study was to test for associations between county-level COVID-19 mortality, pre-pandemic county-level excessive drinking, and county rurality.

### Method

We used national COVID-19 data from the New York Times to calculate county-level case fatality rates (n = 3,039 counties and county equivalents; October 1 –December 31, 2020) and other external county-level data sources for indicators of rurality and health. We used beta regression to model case fatality rates, adjusted for several county-level population characteristics. We included a multilevel component to our model and defined state as a random intercept. Our focal predictor was a single variable representing nine possible combinations of low/mid/high alcohol misuse and low/mid/high rurality.

### Results

The median county-level COVID-19 case fatality rate was 1.57%. Compared to counties with low alcohol misuse and low rurality (referent), counties with high levels of alcohol and

data). County-level indicators of health and demographics used in this study are free and publicly available online through the Robert Wood Johnson County Health Rankings and Roadmaps (https://www.countyhealthrankings.org/explore-health-rankings/rankings-data-documentation). County-level scores for the Index of Relative Rurality used in this study are free and publicly available online through Purdue University's Department of Agricultural Economics (https://purr.purdue.edu/publications/2960/1).

**Funding:** The authors received no specific funding for this work.

**Competing interests:** The authors have declared no competing interests exist.

mid ($\beta$ = -0.17, p = 0.008) or high levels of rurality ($\beta$ = -0.24, p<0.001) demonstrated significantly lower case fatality rates.

## Conclusions

Our findings highlight the intersecting roles of county-level alcohol consumption, rurality, and COVID-19 mortality.

## Introduction

COVID-19, the disease caused by the novel coronavirus SARS-CoV-2, is an ongoing global pandemic. At this writing (January, 2021), the US has seen more cases than any other country, with more than 20 million reported cases and surpassing 350,000 deaths [1]. Co-occurring and pre-existing health conditions exacerbate susceptibility to and the severity of COVID-19 infections [2]. While co-occurring health conditions have received considerable attention, particularly in relation to racial/ethnic disparities [3, 4], there has been less attention paid to behavioral and geographic risk factors associated with COVID-19 outcomes, such as alcohol use and rurality.

Levels of alcohol use and related problems vary across the US. The majority of Americans (73%) report drinking alcohol in the past year, and 13% meet the clinical criteria for a past-year alcohol use disorder [5]. However, alcohol use is not distributed evenly. Previous research has identified distinct regional differences in alcohol consumption [6], and ongoing public health surveillance finds varying levels of hazardous drinking by state [7].

Importantly, moderate to chronic heavy drinking suppresses immune responses and can increase susceptibility to and the seriousness of infectious and respiratory diseases, such as pneumonia, tuberculosis, and acute respiratory stress syndromes [8–10]. This same alcohol-induced immunosuppression may also increase the severity of COVID-19 infections at a population level. Given the effects of alcohol on infectious disease progression, high levels of county-level alcohol use preceding the COVID-19 pandemic are a particularly concerning population health problem.

Many burdens of heavy alcohol use can be exacerbated by other factors. For example, residence in more rural communities is an important social and environmental determinant of health and is associated with lower life expectancy [11], higher rates of all-cause morbidity and mortality [12], and multiple intersecting barriers in access to healthcare [13]. Roughly 60 million people, or nearly one in five Americans, live in a rural area [14].

Rural counties and towns have faced considerable challenges in dealing with the COVID-19 pandemic, partly attributable to weakened health care infrastructure, health care provider shortages, lower socio-economic status, and higher proportions of aging residents [15–17]. In the summer of 2020, the incidence rate of COVID-19 infections began to increase faster in rural areas, following the initial outbreaks in larger and more metropolitan urban centers [18]. Case fatality rates have also been shown to be higher in rural counties [19], rural states [20], and in rural counties with larger proportions of Black and Hispanic populations [21].

Furthermore, alcohol use and misuse vary by rurality. Rural populations are more likely than their urban counterparts to abstain from alcohol, but among non-abstainers, rates of alcohol use disorders (AUD) are higher in more rural areas [22]. Better understanding the association between county-level alcohol use and rurality will inform efforts to mitigate health inequities across regions.

To our knowledge, no studies have investigated the intersection between alcohol misuse, rurality, and COVID-19. This study was designed to measure disparities in COVID-19 infection severity at the county population level, which we operationalized as case fatality rates. We considered two important determinants of population health–excessive alcohol consumption and rurality–and their relationship to case fatality rates at the county level. Although largely an exploratory study, we expected to see higher burdens and more disparities in rural areas.

## Materials and methods

### Data source and sample

We used the New York Times' COVID-19 data repository (https://github.com/nytimes/covid-19-data) to calculate case fatality rates in US counties with at least one COVID-19 death reported between October 1, 2020 and December 31, 2020. We restricted our sample to the last quarter of 2020 because this was the most recent data available at the time of analysis, and to highlight relationships during a time of particularly high case and death rates across the US. Case fatality rates were calculated by dividing the total number of deaths by the total number of confirmed cases, indicating the percentage of individuals with COVID-19 who died. Case fatality rates help to illustrate disproportionate mortality burdens between populations or geographies where the case rates may be similar but underlying social, economic, and health conditions differ. Furthermore, case fatality rates allow for more appropriate comparison between counties with substantial variation in population size.

### Variables

Our focal predictor was a single combined variable representing levels of pre-pandemic county-level excessive alcohol consumption and rurality. The alcohol component was sourced from the Robert Wood Johnson Foundation's 2018 County Health Rankings [23] and merged to the analytic dataset by county. County-level excessive alcohol consumption was defined as the percentage of adults who reported exceeding recommended drinking limits, which included: 1) four or more alcoholic drinks consumed on a single occasion in the past month for women and five or more alcoholic drinks on a single occasion in the past month for men (i.e., binge drinking), and/or 2) more than one drink on average per day for women, and more than two drinks on average per day for men (i.e., exceeding recommended daily limits). We used Waldorf and Kim's Index of Relative Rurality (IRR) to define county-level rurality [24]. The IRR is a threshold-free measurement of rurality at the county level, where greater scores indicate greater rurality. The IRR takes into account population size, population density, remoteness, and built-up infrastructure as a percentage of total land area. Our final analytic variable included nine possible combinations of low/mid/high alcohol consumption and low/mid/high rurality. The low/mid/high cutoffs for both variables were based on tertiles, resulting in nine categorical levels of 1) low alcohol use and low rurality, 2) low alcohol use and mid rurality, 3) low alcohol use and high rurality, 4) mid alcohol use and low rurality, 5) mid alcohol use and mid rurality, 6) mid alcohol use and high rurality, 7) high alcohol use and low rurality, 8) high alcohol use and mid rurality, and 9) high alcohol use and high rurality.

We included several county-level variables in our model that likely confound the relationships between excessive alcohol consumption, rurality, and COVID-19 case fatality rates. All covariates (prevalence of smoking, obesity, diabetes, older age, race/ethnicity, and unemployment) were sourced from the County Health Rankings data. Smoking was defined as the percent of a county's adults who reported currently smoking tobacco every day or on most days, and who have smoked at least 100 cigarettes in their lifetime. Older age was defined as the percent of a county's population aged 65 years and older. Obesity was defined as the percentage of

a county's adults, aged 20 years and older, who report a body mass index of 30 kg/m$^2$ or greater. Diabetes was defined as the percentage of a county's adults, aged 20 years and older, who have been diagnosed with diabetes. Race/ethnicity was represented by the percentage of a county made up of self-identified nonwhite residents. Finally, unemployment was defined as the percentage of a county's civilian labor force, aged 16 and older, who were unemployed but seeking work during the past week.

Our final analytic sample included 3,039 counties and county equivalents in the US that had at least one death recorded and for which we had complete data for all study variables, (i.e., matching FIPS codes between the New York Times and County Health Rankings datasets), representing 96% of all 3,143 US counties and county equivalents.

## Analysis

We used SAS software for all analyses (Version 9.4) [25]. We described the percentage of counties that fall into each level of our categorical predictor, and the median, minimum, and maximum values for all continuous study variables. We reported the IRR scores and case fatality rates for the ten counties with the lowest percentage of excessive drinking and for the ten counties with the highest percentage of excessive drinking. We also developed an original, county-level US map to illustrate the geographic distribution of excessive alcohol consumption and rurality using Mathematica software.

We used beta regression to model case fatality rates. Beta regression is a more appropriate framework than linear regression to model outcomes that are proportions or rates, such as case fatality rates [26]. We considered three separate structures for our model and selected the final model based on how well the model fit the data, as indicated by the lowest Akaike Information Criterion (AIC) value. All candidate models were adjusted for county-level indicators of smoking, obesity, diabetes, age, race/ethnicity, and unemployment. Candidate model #1 used a non-structural, fixed effects design (AIC = -19337). Candidate model #2 included state as a random effect, under the assumption that counties clustered within one state are likely more similar to each other than to counties in other states (AIC = -19845) [27]. Finally, candidate model #3 included state defined as a random effect and accounted for the structural spatial autocorrelation between counties. This model incorporated a Gaussian correlation structure using geographic county centroids (AIC = -19843) [28]. Candidate model #2 was selected as the final analytic model because it demonstrated the lowest AIC value and the best fit to the data. As a multilevel model, we reported the Intraclass Correlation Coefficient (ICC), which is an indicator of how much of the total variation in county case fatality rates is accounted for by the state. Finally, to model our nine-level categorical focal predictor, we included eight dummy variables and used low alcohol use/low rurality as the referent group.

We interpreted study results using a significance threshold of $\alpha = 0.05$. We also identified no problematic multicollinearity between the study variables, defined as a Pearson correlation coefficient greater than 0.80.

## Results

County population characteristics are reported in Table 1. The median county-level COVID-19 case fatality rate was 1.57% (range 0.01% - 16.08%) (Table 2). The high upper limit of this range was occupied by Kenedy County, Texas (16.08%; 184 deaths out of 1,144 reported cases), and the second highest was Franklin County, Massachusetts (11.45%; 7,003 deaths out of 61,156 reported cases). Among all counties, 7.15% (n = 217) were at levels of low alcohol and low rurality, and 10.46% (n = 318) were at levels of high alcohol and high rurality. Eight out of the ten counties with the lowest percentages of excessive drinking were in the US South

**Table 1. County characteristics (n = 3,039 counties).**

| Variables | Median | Min, Max |
|---|---|---|
| Percentage of a county's adult population who reported smoking every day or most days | 17.03 | 5.91, 41.49 |
| Percentage of a county's adult population with a BMI of 30 or higher | 33.20 | 12.40, 57.70 |
| Percentage of a county's adult population with diagnosed diabetes | 11.70 | 1.90, 34.10 |
| Percentage of a county's population aged 65 years and older | 18.83 | 4.83, 57.58 |
| Percentage of a county's population that is nonwhite | 16.76 | 2.11, 97.31 |
| Percentage of a county's civilian labor force, aged 16 and older, that is unemployed but seeking work | 3.88 | 1.30, 18.09 |

(Table 3). The ten counties with the lowest alcohol levels had a combined average IRR score of 0.55 (more rural) and an average case fatality rate of 2.31%. Conversely, all ten counties with the highest levels of excessive alcohol use were in Wisconsin, with a combined average IRR score of 0.45 (less rural) and an average case fatality rate of 0.65%. Counties with both high alcohol and high rurality tended to be clustered in the upper Midwest, parts of the mountain West, and in the states of Nevada and Alaska (Fig 1).

In our fitted model, compared to counties with low alcohol use/low rurality (referent), counties with mid levels of alcohol use and high levels of rurality ($\beta$ = -0.14, p = 0.019), as well as counties with high levels of alcohol use and mid ($\beta$ = -0.17, p = 0.008) and high levels of rurality ($\beta$ = -0.24, p<0.001), demonstrated significantly lower case fatality rates (Table 4). County prevalence of smoking, obesity, and unemployment was not associated with case fatality rate; however, county prevalence of diabetes, older adults, and non-white residents were each positively associated with greater case fatality rates. The ICC value demonstrated that roughly 4% of the variability of case fatality rates between states was attributed to county clustering within states (p<0.001).

## Discussion

To our knowledge, this is the first study to examine county-level relationships between excessive alcohol consumption before the pandemic, rurality, and COVID-19 mortality. Notably,

**Table 2. COVID-19 case fatality rates by county-level alcohol use and rurality (n = 3,039 counties).**

| Variables | n | % | Case fatality rate median (min, max) |
|---|---|---|---|
| *Focal predictor* | | | |
| Low alcohol use counties | | | |
| Low rurality | 217 | 7.15 | 1.84 (0.17, 8.76) |
| Mid rurality | 420 | 13.82 | 1.77 (0.14, 8.99) |
| High rurality | 368 | 12.11 | 1.72 (0.04, 16.08) |
| Mid alcohol use counties | | | |
| Low rurality | 352 | 11.58 | 1.68 (0.01, 7.24) |
| Mid rurality | 301 | 9.90 | 1.71 (0.01, 7.60) |
| High rurality | 352 | 11.58 | 1.65 (0.02, 8.30) |
| High alcohol use counties | | | |
| Low rurality | 433 | 14.25 | 1.48 (0.07, 11.45) |
| Mid rurality | 278 | 9.15 | 1.25 (0.01, 5.62) |
| High rurality | 318 | 10.46 | 1.20 (0.01, 9.30) |

**Table 3. Characteristics of top 10 and lowest 10 counties by county-level excessive drinking.**

| County | State | Excessive drinking[a] (%) | IRR score | Case fatality rate (%) |
|---|---|---|---|---|
| *Lowest ten counties* | | | | |
| Utah | UT | 7.81 | 0.39 | 0.29 |
| Clay | GA | 9.32 | 0.59 | 2.09 |
| Jefferson | MS | 9.49 | 0.57 | 3.34 |
| Holmes | MS | 9.50 | 0.54 | 4.79 |
| Piete | UT | 9.55 | 0.65 | 2.25 |
| Greene | AL | 9.57 | 0.57 | 4.10 |
| Perry | AL | 9.68 | 0.56 | 1.03 |
| McDowell | WV | 9.71 | 0.53 | 0.30 |
| Humphreys | MS | 9.80 | 0.55 | 3.53 |
| Quitman | MS | 9.81 | 0.56 | 1.36 |
| *Top ten counties* | | | | |
| Portage | WI | 27.32 | 0.48 | 0.80 |
| La Crosse | WI | 27.33 | 0.42 | 0.44 |
| Calumet | WI | 27.41 | 0.46 | 0.59 |
| Outagamie | WI | 27.54 | 0.41 | 0.81 |
| Dunn | WI | 27.61 | 0.50 | 0.39 |
| Dodge | WI | 27.92 | 0.47 | 0.91 |
| St. Croix | WI | 27.96 | 0.47 | 0.48 |
| Dane | WI | 28.22 | 0.38 | 0.37 |
| Washington | WI | 28.25 | 0.41 | 0.83 |
| Pierce | WI | 28.62 | 0.50 | 0.84 |

[a] Excessive drinking was defined as past-month binge drinking (≥4 drinks on a single occasion for women; ≥5 drinks on a single occasion for men) or exceeding recommended daily limits (>1 drink per day for women; >2 drinks per day for men).

results indicated a joint contribution of alcohol use patterns and level of rurality. Case fatality rates were generally lowest in more rural counties; compared to low alcohol use/low rurality counties, rates were lower in mid alcohol use/high rurality, high alcohol use/mid rurality and high alcohol use/high rurality counties in particular. Our findings highlight the importance of considering both behavioral and environmental determinants of COVID-19 outcomes.

Our findings did not support our hypothesis of higher burdens and more disparities in rural areas. Instead, we found lower fatality burdens in some mid and high rural counties. In rural settings, other factors besides alcohol use may be more important drivers of COVID-19 fatality, such as a county's age distribution, racial/ethnic distribution, and unemployment levels. Some rural areas may offer protection against the effect of alcohol on mortality. Transmission may be less efficient than in larger and more concentrated urban populations, and drinking norms in rural areas may include smaller and less frequent group meetings in bars or indoor environments, or higher levels of drinking in isolation. Future studies that incorporate longitudinal changes in alcohol use behaviors could improve our understanding of rural and urban differences in health behavior and substance misuse at multiple time points before, during, and following the pandemic.

## Potential limitations

These results should be considered in light of several potential limitations. County-level excessive drinking may have been misestimated as individual survey responses about drinking

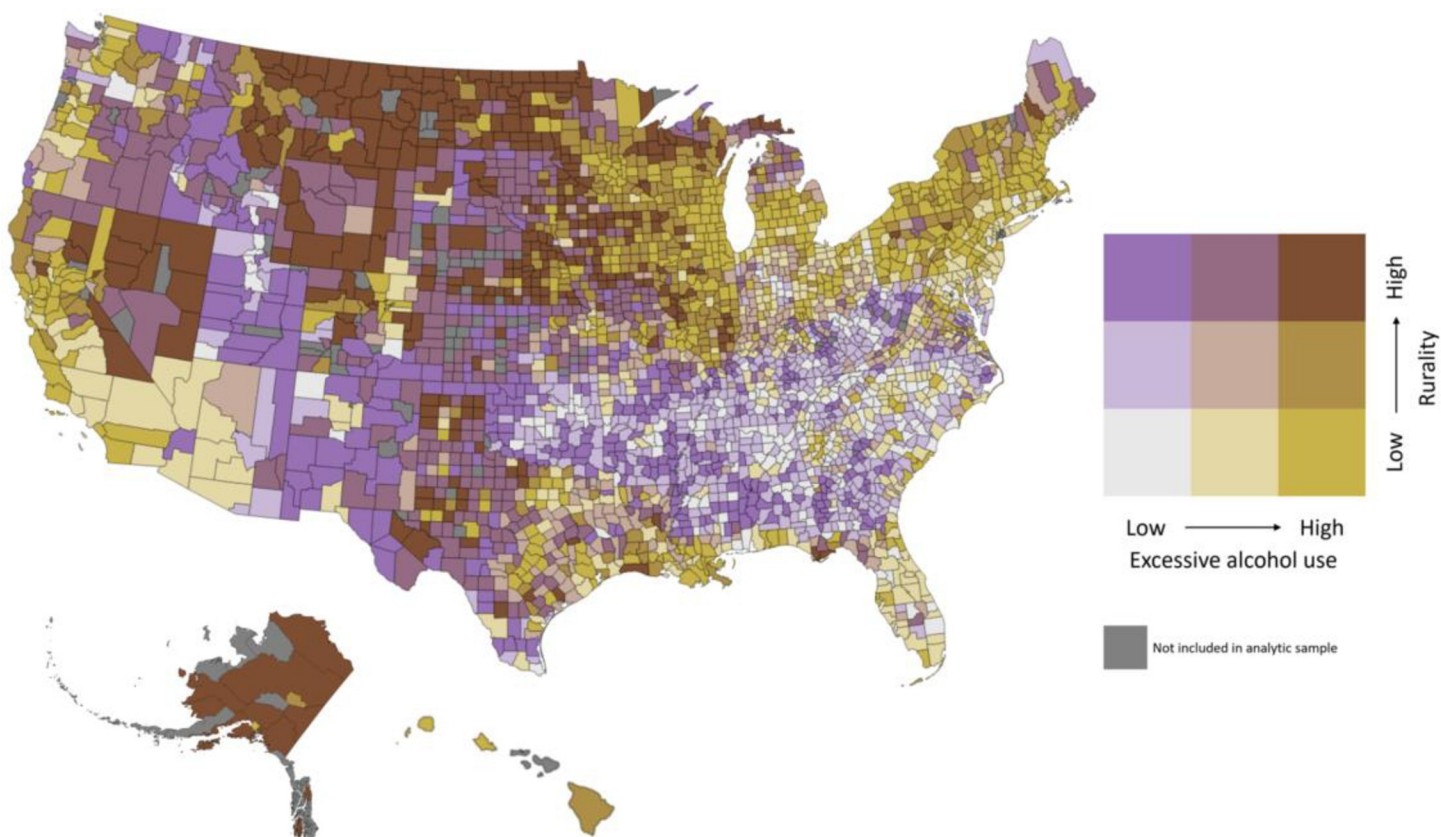

**Fig 1. US county map of levels of excessive alcohol consumption and rurality.** Original map was created by the study team with Mathematica software.

**Table 4. Multivariate beta regression modeling case fatality rate (n = 3,039 counties).**

| Variables | *β* | SE | p |
|---|---|---|---|
| *Focal predictor* | | | |
| Low alcohol use counties | | | |
| Low rurality | Ref. | Ref. | Ref. |
| Mid rurality | -0.01 | 0.04 | 0.982 |
| High rurality | -0.07 | 0.05 | 0.138 |
| Mid alcohol use counties | | | |
| Low rurality | -0.06 | 0.05 | 0.197 |
| Mid rurality | -0.07 | 0.06 | 0.207 |
| High rurality | -0.14 | 0.06 | 0.019 |
| High alcohol use counties | | | |
| Low rurality | -0.09 | 0.06 | 0.110 |
| Mid rurality | -0.17 | 0.07 | 0.008 |
| High rurality | -0.24 | 0.07 | <0.001 |
| *Covariates* | | | |
| Percentage of a county's adult population who reported smoking every day or most days | -0.01 | 0.01 | 0.671 |
| Percentage of a county's adult population with a BMI of 30 or higher | 0.01 | 0.01 | 0.259 |
| Percentage of a county's adult population with diagnosed diabetes | 0.01 | 0.01 | 0.032 |
| Percentage of a county's population aged 65 years and older | 0.02 | 0.01 | <0.0001 |
| Percentage of a county's population that is nonwhite | 0.01 | 0.01 | <0.0001 |
| Percentage of a county's civilian labor force, aged 16 and older, that is unemployed but seeking work | 0.01 | 0.01 | 0.552 |

Intraclass correlation coefficient = 0.037, p<0.001.

habits may be subject to social desirability biases [29]. Such underreporting would decrease our ability to detect an association. Second, as a secondary data analysis, some variables were not available, such as asymptomatic COVID-19 cases, COVID-19 testing capacity, relevant county-level comorbidities (i.e., chronic lung disease or asthma, liver disease, serious heart conditions, and other diseases affecting immune response), and county healthcare characteristics (i.e., available ventilators and provider shortage areas). Finally, as an ecological study using county-level data, no individual-level information was available. As such, there is no way to identify the demographic characteristics of the population that made up the cases and deaths. Future studies that evaluate relationships between individual-level alcohol use and COVID-19 mortality outcomes would contribute substantially to the literature around behavioral health and infectious disease outbreaks. As this study was undertaken in the midst of the COVID-19 pandemic, our pattern of results should be confirmed as more data accumulate. Nevertheless, our findings address a gap in knowledge about the association of excessive alcohol consumption, rurality, and COVID-19 outcomes.

## Conclusion

Our findings demonstrate lower COVID-19 fatality burdens in some counties, in particular counties with mid/high excessive alcohol use and counties with mid/high levels of rurality. These findings were contrary to our hypothesis and highlight the intersecting roles that excessive alcohol consumption and geography play in baseline risk for COVID-19 outcomes at the community and population levels.

## Author Contributions

**Conceptualization:** George Pro, Paul A. Gilbert, Julie A. Baldwin, Clare C. Brown, Nickolas Zaller.

**Data curation:** George Pro.

**Formal analysis:** George Pro, Sean Young.

**Investigation:** George Pro.

**Methodology:** George Pro, Paul A. Gilbert, Julie A. Baldwin, Clare C. Brown, Sean Young, Nickolas Zaller.

**Project administration:** George Pro.

**Software:** George Pro.

**Supervision:** Nickolas Zaller.

**Visualization:** George Pro, Sean Young.

**Writing – original draft:** George Pro.

**Writing – review & editing:** George Pro, Paul A. Gilbert, Julie A. Baldwin, Clare C. Brown, Sean Young, Nickolas Zaller.

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
