## [Decision Letter · Decision Letter 0]

29 Mar 2021

PONE-D-21-01356

Multilevel modeling of county-level excessive alcohol use, rurality, and COVID-19 case fatality rates in the US

PLOS ONE

Dear Dr. Pro,

Thank you for submitting your manuscript to PLOS ONE. After careful consideration, we feel that it has merit but does not fully meet PLOS ONE’s publication criteria as it currently stands. Therefore, we invite you to submit a revised version of the manuscript that addresses the points raised during the review process.

We look forward to receiving your revised manuscript.

Kind regards,

Antonio Palazón-Bru, PhD

Academic Editor

PLOS ONE

Journal Requirements:

2. Please refer to the specific statistical analyses performed as well as any post-hoc corrections to correct for multiple comparisons. If these were not performed please justify the reasons. Please refer to our statistical reporting guidelines for assistance (https://journals.plos.org/plosone/s/submission-guidelines.#loc-statistical-reporting). Additionally, please ensure you have thoroughly discussed any potential limitations of this study within the Discussion.

4. We note that Figure 1 in your submission contain map images which may be copyrighted. All PLOS content is published under the Creative Commons Attribution License (CC BY 4.0), which means that the manuscript, images, and Supporting Information files will be freely available online, and any third party is permitted to access, download, copy, distribute, and use these materials in any way, even commercially, with proper attribution. For these reasons, we cannot publish previously copyrighted maps or satellite images created using proprietary data, such as Google software (Google Maps, Street View, and Earth). For more information, see our copyright guidelines: http://journals.plos.org/plosone/s/licenses-and-copyright.

4.1.    You may seek permission from the original copyright holder of Figure 1 to publish the content specifically under the CC BY 4.0 license. 

4.2.    If you are unable to obtain permission from the original copyright holder to publish these figures under the CC BY 4.0 license or if the copyright holder’s requirements are incompatible with the CC BY 4.0 license, please either i) remove the figure or ii) supply a replacement figure that complies with the CC BY 4.0 license. Please check copyright information on all replacement figures and update the figure caption with source information. If applicable, please specify in the figure caption text when a figure is similar but not identical to the original image and is therefore for illustrative purposes only.

Reviewers' comments:

Reviewer's Responses to Questions

**Comments to the Author**

1. Is the manuscript technically sound, and do the data support the conclusions?

Reviewer #1: Yes

Reviewer #2: No

2. Has the statistical analysis been performed appropriately and rigorously? 

Reviewer #1: Yes

Reviewer #2: Yes

3. Have the authors made all data underlying the findings in their manuscript fully available?

Reviewer #1: Yes

Reviewer #2: Yes

4. Is the manuscript presented in an intelligible fashion and written in standard English?

Reviewer #1: Yes

Reviewer #2: Yes

5. Review Comments to the Author

Reviewer #1: Summary:

This study wanted to determine if county COVID-19 case fatality rate could be predicted by the 9-level grouping of alcohol use (low, medium, high) and rurality (low, medium, high) levels, while controlling for state, and other health factors (such as smoking, age, diabetes, etc.). They found that, contrary to their exploratory hypothesis, there were lower case fatality rates in higher alcohol-using and more rural counties than the baseline (low alcohol, low rural).

This manuscript is well-written and covers a relevant topic. Furthermore, the model design is well-founded and had appropriate controls and justification. Finally, the results were very interesting, as they were the opposite of what is expected and had a nice bevy of detail. The major weakness of the manuscript was an incorrect assertion in the discussion/conclusion about the burden of COVID-19 which was not supported by any results. A related weakness is the call-to-action statement, which was in no way justified by the results.

Specific Areas of Improvement:

Abstract:

Major:

Page 2, Line 23-27: Your conclusion has a call-to-action that includes alcohol prevention/treatment in the pandemic response plan. While mitigating alcohol abuse is a good in of itself, increasing efforts based on the pandemic is not supported by your results. Counties with higher alcohol levels had lower case fatality rates. You shouldn’t include such a call-to-action unless you can convincingly demonstrate why, despite the contrary results, that such a response is warranted, and I don’t see how you could.

Minor:

Page 2, Lines 15-17: consider adding ‘while controlling for confounding factors’ to the model description.

Page 2, Line 21: p-values should have consistent significant digits (I typically see 3-4 as the preferred number in other manuscripts).

Introduction:

Major:

none

Minor:

none

Methods:

Major:

none

Minor:

Page 5, Line 5: For the COVID-19 deaths, you used only the most recent 3 months. Is there a reason why? That time period, while likely a reasonable approach, should be justified.

Results:

Major:

none

Minor:

Page 12, Lines 2-4, 5: As noted in the abstract, the p-values should have consistent significant digits.

Figures and Tables:

Major:

none

Minor:

Table 2: The header for the binge drinking column is much longer than the other headers. Perhaps shorten the header for binge drinking and give the additional information into the table description [example for revised header: Excess drinking (%)].

Table 3: For the second column header, change the B to β to match the format from the text portion of the results. For the p-value column, again set the p-values to consistent significant digits.

Discussion:

Major:

Page 14, Lines 4-7: You write that the burden of higher case fatality rates lay predominately in mid alcohol use/mid rural and high alcohol use/high rural counties. This is not what the results show. High alcohol use/high rural counties had the lowest case fatality rates, while mid alcohol use/mid rural had lower rates (though not significant) to the referent (low alcohol/low rural). These two sentences need to be revised to reflect what the results found.

Page 14, Lines 19-23; Page 15, Lines 1-7: You write that alcohol consumption during self-isolation poses a substantial public health challenge. However, none of your results provide any evidence for this claim. The three categories of counties with high alcohol had the lowest case fatality rates (1.48%, 1.25%, and 1.20%, from Table 1). Furthermore, even looking at the top 10 highest and lowest alcohol abuse counties, the average case fatality rates are far lower in in the highest (0.646%) compared to the lowest (2.308%) (values calculated by averaging the case-fatality-rate values in Table 2). As noted in the abstract, while mitigating alcohol abuse is a good in of itself, increasing efforts based on the pandemic is not supported by your results. This whole section needs to be dropped or revised to justify such action despite contrary results (which I do not see how you could).

Page 15, Lines 21-22; Page 16, Line 1: This sentence needs to be corrected in the same way as the Page 14, Lines 4-7.

Page 16, Lines 3-5: This call-to-action sentence, like the section of Page 14, Lines 19-23; Page 15, Lines 1-7, has no support from your results. It should not be included.

Minor:

Page 14, Line 5: the word ‘lied’ should be ‘lay’

Page 15, Line 8: The biggest potential limitation is that this is an ecological study (i.e., the data was aggregated at the county level, so individual metrics were not available). You cannot know if the population that made up the cases and deaths have the same demographics as the other variables (rurality, alcohol use, etc.). While this may well be an obvious statement, it is good to mention it since you’ve included a section on potential limitations.

Page 15, Line 20: For your conclusion, because of the following the facts: 1) the results were contrary to expectations, 2) it was an ecological study, and 3) you’re trying to give alcohol prevention recommendations, it may be a good idea to outline a future experiment that would test the link between mortality and alcohol use on an individual level.

Specific Areas of Achievement:

Abstract:

Objective, methods, and results sections were clear and concise, presenting the reader with an understandable summary of your project.

Introduction:

Page 4, Lines 20-23: The statement of the project and the hypothesis are good additions.

Methods:

Page 7, Lines 10-23; Page 8, Lines 1-3: the description of the model building and selection was very clear and well justified. I thoroughly enjoyed reading this section.

Results:

Short and to the point. It conveyed the major results while letting the details reside in the appropriate figures and graphs.

Figures and Tables:

Figure 1 was particularly well made. Tables were clear and informative.

Discussion:

While the call-to-action was no appropriate for the results found, I do like the attempt to translate results into action.

Reviewer #2: This paper explored the relationship between county-level excessive alcohol consumption, rurality and COVID-19 case-fatality rate using multilevel beta regression analyses. The authors make a compelling argument for the pathway through which a higher prevalence of excessive alcohol consumption may affect COVID-19 fatality. They also make a compelling argument in the discussion section about potential reasons for the reversed associations between the outcome and focal predictor. The paper is methodologically sound. However, as indicated in the feedback below, some of the discussion and conclusion points are in contrast to the results shown in the tables. Minor revisions are suggested.

Tables.

Table 1. For ease of readability, the authors should consider providing summary statistics in table 1 for only the covariates (second part of table 1) and having the first part showing case-fatality rates for values of the focal predictor as a separate table.

Table 3. The low alcohol mid rurality coefficient is reported as 0.00, the coefficient should be rounded to a non-zero value. Same applies to “percentage of a county’s adult population with a BMI of 30 or higher” covariate and several standard errors.

Discussion.

Page 14 line 4-6 “We found that the burden of higher case fatality rates lied predominately in mid alcohol use/mid rural and high alcohol use/high rural counties.” However, table 3 shows that the mid alcohol /mid rural was not statistically significant and also did not have the largest effect size (the mid alcohol/high rural and high alcohol/mid rural were significant).

Page 15 line 2 “Given indications that alcohol use is associated with higher COVID-19 mortality in some areas,…” this is in contrast to the index study finding and the authors do not provide any citation to support the statement.

Page 15 line 21-page 16 line 1 “our findings demonstrate disparate COVID-19 fatality burdens in some counties, in particular counties with mid/high excessive alcohol use and counties with mid/high levels of rurality.” This again is not supported by the study findings. Table 1 and table 2 indicate that the case fatality burden is more for the low alcohol/low rural counties.

6. PLOS authors have the option to publish the peer review history of their article (what does this mean?). If published, this will include your full peer review and any attached files.

Reviewer #1: **Yes: **Mark R Williamson

Reviewer #2: No

---

## [Author Response · Author response to Decision Letter 0]

7 May 2021

Thank you to the academic editor and our two reviewers for the feedback and constructive criticism of our manuscript. The changes we made in response to the reviews have strengthened the paper overall, with a focus on keeping our discussion in line with our reported findings and refraining from making recommendations that do not stem directly from our results. Below we have summarized some of the reviews and combined others that were noted by both reviewers. We have provided one revised version with tracked changes, and another clean version with changes accepted.

General feedback from the academic editor

1) We note that Figure 1 in your submission contains map images which may be copyrighted. We cannot publish previously copyrighted maps or satellite images created using proprietary data. We require you to either (1) present written permission from the copyright holder to publish these figures specifically under the CC BY 4.0 license, or (2) remove the figures from your submission.

We have communicated with our PLOS publishing editor, Theodore Peng, about our map. This original map was created with Mathematica software by the study team using the same data from the study analyses. The map image is not copyrighted.

We have included additional information about the creation of the map in the methods section and added a caption to the figure specifying that the map was created by the study team using Mathematica.

Abstract

2) Page 2, Line 23-27: Your conclusion has a call-to-action that includes alcohol prevention/treatment in the pandemic response plan. While mitigating alcohol abuse is a good in of itself, increasing efforts based on the pandemic is not supported by your results. Counties with higher alcohol levels had lower case fatality rates. You shouldn’t include such a call-to-action unless you can convincingly demonstrate why, despite the contrary results, that such a response is warranted, and I don’t see how you could. (Reviewer 1)

We agree that this call-to-action is beyond the scope of the study findings. We have removed this statement from the abstract as well as throughout the discussion section.

3) Page 2, Lines 15-17: consider adding ‘while controlling for confounding factors’ to the model description. (Reviewer 1)

We have included a statement in the abstract clarifying that our model was adjusted for relevant county-level population characteristics.

4) Page 2, Line 21: p-values should have consistent significant digits (I typically see 3-4 as the preferred number in other manuscripts). (Reviewer 1)

We have reviewed the PLOS reporting guidelines for p-values, and have reported all p-values to three decimal places throughout the manuscript.

Methods

5) Page 5, Line 5: For the COVID-19 deaths, you used only the most recent 3 months. Is there a reason why? That time period, while it is likely a reasonable approach, should be justified. (Reviewer 1)

We have clarified our justification for using this date range, including that it was a time of particularly high case and death rates across the US, and was also the most recent data available at the time of analysis. 

Results

6) Table 1. For ease of readability, the authors should consider providing summary statistics in table 1 for only the covariates (second part of table 1) and having the first part showing case-fatality rates for values of the focal predictor as a separate table. (Reviewer 2)

Thank you for helping us improve the readability of our tables. We have revised our tables to present only summary statistics for county population characteristics in Table 1, then moved the case fatality rate information to a new Table 2. 

7) Table 2: The header for the binge drinking column is much longer than the other headers. Perhaps shorten the header for binge drinking and give the additional information into the table description [example for revised header: Excess drinking (%)]. (Reviewer 1)

We shortened the column header to the suggested “Excessive drinking (%)”, which helped clean up this table and keep column headers small and similarly sized.

8) Table 3: For the second column header, change the B to β to match the format from the text portion of the results. (Reviewer 1)

This has been updated in Table 3.

9) Table 3. The low alcohol mid rurality coefficient is reported as 0.00, the coefficient should be rounded to a non-zero value. Same applies to “percentage of a county’s adult population with a BMI of 30 or higher” covariate and several standard errors. (Reviewer 2)

We have rounded the small coefficient and standard error values to the nearest non-zero value using two decimal places.

Discussion

10) Page 14, Lines 4-7: You write that the burden of higher case fatality rates lay predominately in mid alcohol use/mid rural and high alcohol use/high rural counties. This is not what the results show. High alcohol use/high rural counties had the lowest case fatality rates, while mid alcohol use/mid rural had lower rates (though not significant) to the referent (low alcohol/low rural). These two sentences need to be revised to reflect what the results found. (Reviewer 1)

Page 14 line 4-6 “We found that the burden of higher case fatality rates lied predominately in mid alcohol use/mid rural and high alcohol use/high rural counties.” However, table 3 shows that the mid alcohol /mid rural was not statistically significant and also did not have the largest effect size (the mid alcohol/high rural and high alcohol/mid rural were significant). (Reviewer 2)

We appreciate that these mistakes were pointed out. We have clarified our findings and the direction of significant associations in this first discussion paragraph.

11) Page 14, Lines 19-23; Page 15, Lines 1-7: You write that alcohol consumption during self-isolation poses a substantial public health challenge. However, none of your results provide any evidence for this claim. This whole section needs to be dropped or revised to justify such action despite contrary results (which I do not see how you could). (Reviewer 1)

We agree that this paragraph expands the discussion beyond the study findings. In order to keep our discussion aligned with the reported findings – and not venture into a call-for-action or recommendations for treatment interventions – we have removed this paragraph.

Limitations

12) Page 15, Line 8: The biggest potential limitation is that this is an ecological study. You cannot know if the population that made up the cases and deaths have the same demographics as the other variables. While this may well be an obvious statement, it is good to mention it since you’ve included a section on potential limitations. (Reviewer 1)

We have added a short paragraph outlining the limitations of ecological study designs, as well as a recommendation for future observational studies identifying pathways between behavioral health and COVID-19 outcomes at the individual level.

Conclusion

13) Page 15, Line 20: For your conclusion, because of the following the facts: 1) the results were contrary to expectations, 2) it was an ecological study, and 3) you’re trying to give alcohol prevention recommendations, it may be a good idea to outline a future experiment that would test the link between mortality and alcohol use on an individual level. (Reviewer 1)

Please see our response to Comment #12, above. We have drawn attention to the limitations of interpreting results from ecological studies, while also making a recommendation for future observational studies that are well suited to identify associations between individual-level characteristics like health behavior and disease outcomes.

14) Page 15 line 21-page 16 line 1 “our findings demonstrate disparate COVID-19 fatality burdens in some counties, in particular counties with mid/high excessive alcohol use and counties with mid/high levels of rurality.” This again is not supported by the study findings. Table 1 and table 2 indicate that the case fatality burden is more for the low alcohol/low rural counties. (Reviewer 2)

We have made efforts throughout the manuscript to correctly report our findings and clarify the direction of associations, including in the conclusion section. We have also removed our recommendations for strengthened alcohol screening and treatment as they are not within the scope of our reported findings.

---

## [Decision Letter · Decision Letter 1]

7 Jun 2021

Multilevel modeling of county-level excessive alcohol use, rurality, and COVID-19 case fatality rates in the US

PONE-D-21-01356R1

Dear Dr. Pro,

We’re pleased to inform you that your manuscript has been judged scientifically suitable for publication and will be formally accepted for publication once it meets all outstanding technical requirements.

Kind regards,

Antonio Palazón-Bru, PhD

Academic Editor

PLOS ONE

Additional Editor Comments (optional):

Reviewers' comments:

Reviewer's Responses to Questions

**Comments to the Author**

1. If the authors have adequately addressed your comments raised in a previous round of review and you feel that this manuscript is now acceptable for publication, you may indicate that here to bypass the “Comments to the Author” section, enter your conflict of interest statement in the “Confidential to Editor” section, and submit your "Accept" recommendation.

Reviewer #1: All comments have been addressed

Reviewer #2: All comments have been addressed

2. Is the manuscript technically sound, and do the data support the conclusions?

Reviewer #1: Yes

Reviewer #2: Yes

3. Has the statistical analysis been performed appropriately and rigorously? 

Reviewer #1: Yes

Reviewer #2: Yes

4. Have the authors made all data underlying the findings in their manuscript fully available?

Reviewer #1: Yes

Reviewer #2: (No Response)

5. Is the manuscript presented in an intelligible fashion and written in standard English?

Reviewer #1: Yes

Reviewer #2: (No Response)

6. Review Comments to the Author

Reviewer #1: Revised version looks good. A strength of this manuscript is the clarity of statistical analysis and the use of a single variable to capture both alcohol usage and rurality into categories.

Reviewer #2: (No Response)

7. PLOS authors have the option to publish the peer review history of their article (what does this mean?). If published, this will include your full peer review and any attached files.

Reviewer #1: **Yes: **Mark R Williamson

Reviewer #2: No

---

## [Editor Report · Acceptance letter]

9 Jun 2021

PONE-D-21-01356R1 

Multilevel modeling of county-level excessive alcohol use, rurality, and COVID-19 case fatality rates in the US 

Dear Dr. Pro:

I'm pleased to inform you that your manuscript has been deemed suitable for publication in PLOS ONE. Congratulations! Your manuscript is now with our production department. 

Kind regards, 

on behalf of

Dr. Antonio Palazón-Bru 

Academic Editor

PLOS ONE